# Newest Therapies for Cholangiocarcinoma: An Updated Overview of Approved Treatments with Transplant Oncology Vision

**DOI:** 10.3390/cancers14205074

**Published:** 2022-10-17

**Authors:** Yuqi Zhang, Abdullah Esmail, Vincenzo Mazzaferro, Maen Abdelrahim

**Affiliations:** 1Department of Medicine, Houston Methodist Hospital, Houston, TX 77030, USA; 2Section of GI Oncology, Department of Medical Oncology, Houston Methodist Cancer Center, Houston, TX 77030, USA; 3Cancer Clinical Trials, Houston Methodist Research Institute, Houston, TX 77030, USA; 4Department of Oncology, University of Milan, 20133 Milan, Italy; 5Gastro-Intestinal Surgery and Liver Transplantation Unit, The Instituto Nazionale Tumori (National Cancer Institute) of Milan, 20133 Milan, Italy; 6Weill Cornell Medical College, New York, NY 14853, USA; 7Cockrell Center for Advanced Therapeutic Phase I program, Houston Methodist Research Institute, Houston, TX 77030, USA

**Keywords:** cholangiocarcinoma, targeted therapy, transplant oncology, immunotherapy

## Abstract

**Simple Summary:**

Cholangiocarcinoma is a relatively rare but deadly disease with traditionally limited treatment options. The disease can be categorized by anatomic location within the biliary tree, with different associated risk factors and molecular profiles. Recent years have seen a burgeoning of targeted therapies that have enhanced survival in subsets of patients with certain mutations. We herein discuss these more recent advances as well as providing an overview of more well-known treatment modalities, with the goal of providing an accessible source for practicing clinicians.

**Abstract:**

A minority of cholangiocarcinoma (CCA) can be cured by surgical intervention (i.e., liver resection (LR) and liver transplantation (LT)). When modern criteria for LT are met, this intervention along with neoadjuvant treatments may achieve unprecedented survival in selected patients. Liver resection is associated with a median overall survival (OS) of 40 months, this number drastically decreases for unresectable advanced cholangiocarcinoma (CCA), which is treated with systemic therapy. The first-line chemotherapy regimen of gemcitabine and cisplatin is associated with a median overall survival of only 11.7 months. Since the Food and Drug Administration (FDA)’s approval of the isocitrate dehydrogenase (*IDH*) 1 inhibitor ivosidenib in August 2021, there has been increasing interest in targeted therapy for CCA patients harboring mutations in fibroblast growth factor receptor (*FGFR*) 2, neurotrophic receptor tyrosine kinase (*NTRK*), B-raf kinase (*BRAF*), and *HER2*. At the same time, immunotherapy with immune checkpoint inhibitors isalso being used in relapsed CCA. This review looks into the most recently completed and ongoing studies of targeted therapy as monotherapy or in combination with chemo- and/or immunotherapy. Whether it is resection, liver transplant, radiotherapy, chemotherapy, immunotherapy, targeted therapy, or any combination of these treatment modalities, great strides are being made to improve outcomes for this challenging disease.

## 1. Introduction

According to the National Cancer Institute, about 2000–3000 new cases of bile duct cancer or cholangiocarcinoma (CCA) are diagnosed in the United States every year [1]. CCA is formally defined as cancer arising from the bile duct epithelium and is further categorized according to anatomical location as intrahepatic (iCCA), perihilar (pCCA, also known as hilar CCA), or distal (dCCA) [2]. The last two categories may be grouped as extrahepatic CCA (eCCA). iCCA arises from the bile ductules and segmental ducts (proximal to the second-order ducts) while pCCA arises from the left, right, and common hepatic ducts. Finally, dCCA arises from the common bile duct (past the insertion of the cystic duct). While also a form of hepatobiliary cancer, gallbladder cancers arise from either the gallbladder or the cystic duct and are beyond the scope of this review.

Most CCAs are adenocarcinomas with rare squamous, adenosquamous, mucinous, or signet ring cell, clear cell, undifferentiated, and lymphoepithelial types [3]. iCCAs are mostly (78%) mass forming, with a minority being periductal-infiltrating or intraductal-growing. The pCCAs and dCCAs are mostly periductal-infiltrating (73%) and described as flat or nodular sclerosing, or intraductal papillary (27%). While precursor lesions such as intraductal papillary neoplasms of the bile duct (IPNBs) have been identified for periductal-infiltrating and papillary types of CCAs, precursors for mass-forming iCCAs have not been found.

To better guide management decisions, there have been continuing efforts to modify the staging system for CCA. Starting from the 7th edition of the AJCC staging system, the classification of iCCA was by clinicopathologic features such as multiple hepatic tumors, regional nodal involvement, and tumor size. In the 8th edition, T1 (solitary tumor without vascular invasion) has been further divided into T1a and T1b according to size (with the latter being >5 cm). Resection of at least six nodes is now recommended for accurate staging [4]. For eCCA, the 7th and 8th editions of the AJCC staging system both consider the extent of liver involvement and distant metastasis, with the depth of tumor invasion as an independent prognostic factor [5,6]. The 8th edition further categorizes regional lymph node involvement according to the number of positive lymph nodes [5,7]. Neither the AJCC nor the Bismuth–Corlette staging system can be used to predict survival or resectability, but the Blumgart system (a preoperative staging system which incorporates radial and longitudinal tumor dimensions) does [7,8]. pCCA is classified into T1–T3 based on the location and extent of bile duct involvement, the presence of portal vein invasion, and hepatic lobe atrophy [9]. While surgery has remained the mainstay of curative therapy and prognosis is still significantly affected by resectability, the last decade has seen significant advancements in systemic therapies. With the discovery of actionable mutations in 40–50% of patients with advanced CCA [10], increasing attention has been turned to targeted therapy for locally advanced, chemo-refractory, and metastatic CCA, which is the focus of this review.

## 2. Surgical Resection and Liver Transplantation

### 2.1. iCCA

Several criteria are considered in determining tumor resectability in CCA. In addition to the location of the tumor, as most of the patients (70–80%) require a major hepatectomy (i.e., ≥3 segments), resectability depends on the amount and quality of the predicted future liver remnant (FLR). Greater than or equal to 25% FLR is acceptable in the normal liver and greater than or equal to 40% in the diseased (e.g., cirrhotic) liver [3,11]. When the threshold FLR is not met, portal vein embolization (PVE) or the accelerated procedure called associating liver partition and portal vein ligation for staged hepatectomy (ALPPS) are sometimes used to promote left lobe hypertrophy. Resection is associated with a median overall survival of 40 months, with 5-month survival estimated at 25–40% [3,12].

The last decade witnessed a scientific revolutionof the transplant oncology field that provides an excellent option to treat patients with iCCA by liver transplant [12,13,14,15,16,17,18,19,20,21,22,23]. Two main scenarios are worth attention in the transplant perspective.

First, evidence supports liver transplantation (LT) in “very early” iCCA (in which the tumor isno more than 2 cm) arising in the context of hepatic cirrhosis and not amenable to liver resection because of poor liver function. A key study in 2016 reported lower recurrence and improved survival after LT in such patients compared to those with more advanced iCCAs [13,24]. Moreover, a recent multicentric experience in France has observed for patients with larger lesions (2–5 cm) a post-LT 5-year OS and RFS comparable to those of patients with tumors ≤2 cm, noting that differentiation rather than size correlated with tumor recurrence on multivariate analysis [25].

Secondly, LT may be justifiable in non-resectable iCCA arising in a non-cirrhotic liver when multimodal treatment strategies encompass optimal patients’ selection based on tumor-related factors (i.e., lymph node metastases, levels of Ca19.9 with variable cutoffs from 100 to 500 U/mL) and on disease control over time by means of systemic and loco-regional treatments. Based on these observations, researchers at the University of California, Los Angeles, have devised a prognostic scoring system that stratifies patients as having low (78% recurrence-free survival rate), intermediate (19%), and high (0%) risk of recurrence which can be used in consideration for a liver transplant with significantly improved outcomes [13,26]. Until 2000, reported survival rates for iCCA after liver transplant have been 50% at 3 years and 20% at 5 years [13]. A recent study by Houston Methodist J.C. Walter Jr. Liver Transplant Center and MD Anderson Cancer Center of sixiCCA patients treated with neoadjuvant chemotherapy and LT showed a 5-year OS of 83.3% (with a 50% recurrence rate). This brought the International Liver Cancer Association to support further clinical trials to investigate the efficacy of neoadjuvant chemotherapy with LT [13,27]. Additionally, locoregional treatments (including trans-arterial chemoembolization: TACE, selective internal radiotherapy: SIRT, and stereotactic external beam radiotherapy: EBRT) have been proposed to complement systemic treatments and improve the patient section in light of liver transplant. A key aspect of such strategies lies in a pre-listing observation time of the patient during which neoadjuvant treatments can be administered in order to sustain tumor control (either response or stability) and assess tumor biology. Similarly, to other indications in transplant oncology, the observation time ahead of transplant and the grade of response to non-surgical neoadjuvant treatments have been shown to be key factors to maximize the survival benefit of LT.

### 2.2. Extrahepatic Cholangiocarcinoma

In addition to the FLR, special considerations for resection of perihilar CCA include the need for preoperative biliary drainage in cases of obstruction and cholangitis. In the absence of cholangitis, drainage may be warranted to increase FLR [3,28]. Additionally, vascular clearance and the ability to reconstruct the biliary tree must be considered. Portal vein resection may sometimes be pursued for R0-resection despite increased short-term mortality [3]. Some patients may also be considered for a liver transplant. The Mayo protocol, which consists of external beam radiation to 45 Gy with continuous infusional 5-fluorouracil followed by intrabiliary radiation (brachytherapy with iridium) and oral capecitabine with ambulatory 5-FU infusion until the time of LT, has resulted in 5-year survival rates of 53–54% in patients with stage I and II disease in two different studies [29,30]. Notably, 5-year recurrence-free survival rate was 65%, supporting the effectiveness of transplant as a treatment modality for perihilar CCA [13,29]. As previously pointed out by our group, conflicting results of transplants among centers based in the U.S. and other countries may also rest on the criteria for resectability [13]. Regardless, the role of transplant in CCA is an area of active investigation, and results of the French TRANSPHIL study (NCT02232932), which compares chemotherapy followed by transplant vs. liver resection in hilar CCA, are eagerly awaited.

## 3. Systemic Therapies

Systemic therapy is recommended for unresectable and metastatic biliary tract cancers, as well as disease with microscopic margins (R1), and positive regional lymph nodes after resection. To guide therapy, additional MSI/MMR, molecular, and TMB testing is recommended.

Adjuvant and neoadjuvant chemotherapy are active topics of research in cholangiocarcinoma. Adjuvant chemotherapy with capecitabine has been recommended for node-positive CCA given the results of the BILCAP study. The study revealed a significant overall survival (51.1 vs. 36.4 months) and relapse-free survival benefit in CCA and gallbladder cancer patients who were treated with capecitabine for 6 months after resection compared to surgery alone [3]. Several studies are still underway to investigate the efficacy of other adjuvant therapies such as S1 (a combination of tegafur, gimeracil and oteracil) (JCOG-1202) and the combination of cisplatin and gemcitabine (ACTICCA-1). Notably, neoadjuvant chemotherapy hasalso been associated with longer overall survival in a select group of patients in a propensity score analysis of patients with stage I–III CCA between 2006–2014 who received both chemotherapy and surgery [31]. While no standard neoadjuvant chemotherapy for CCA is recommended, various treatment modalities including treatment with gemcitabine, 5-FU, etc., as well as radiotherapy, have been shown to convert “potentially resectable” or borderline-resectable CCA to resectable disease [32].

The preferred first-line chemotherapy regimen for advanced cholangiocarcinoma is gemcitabine and cisplatin, based on the randomized, controlled, Phase 3 ABC-02 study [7,33]. This regimen improved median OS (11.7 vs. 8.1 months) and PFS (8 vs. 5 months) by 30% compared to gemcitabine alone [33]. Other treatments are either gemcitabine or fluoropyrimidine-based. Several studies have looked at gemcitabine used in combination with oxaliplatin, albumin-bound (nab-)paclitaxel, and cetuximab, while fluoropyrimidine-based treatments also include cisplatin or oxaliplatin [7]. Median OS and PFS for first-line gemcitabine combined with nab-paclitaxel is similar to that of gemcitabine and cisplatin at 12.4 and 7.7 months, respectively in Phase 2 clinical trial (NCT02181634) [34]. A Phase 2 trial (NCT00747097) of gemcitabine and cetuximab showed similar median OS and PFS of 13.5 (95% CI: 9–31.8) months and 5.8 months (95% CI: 3.6–8.5 months), with the most common side effects being hematological (Grade 3 or 4 more) [35]. There have also been various Phase 2 studies supporting category 2A recommendations for gemcitabine with oxaliplatin or capecitabine, capecitabine with oxaliplatin, and the single agents of fluorouracil, capecitabine, and gemcitabine.

More recently, attention has been turned to triple combinations of chemotherapy. For example, the use of the combination of gemcitabine, cisplatin, and nab-paclitaxel in patients with advanced eCCA, iCCA, and gallbladder cancer in a single-arm Phase 2 trial showed improved median OS (19.2 months (95% CI, 13.2 months to not estimable)) and PFS (11.8 months (95% CI, 6 to 15.6 months)) [36]. Notably, the same study showed that the tumors in 20% of patients who previously had unresectable disease became resectable. With intensified chemotherapy, there have been increased toxicities which may limit widespread implementation, with Grade 3 more higher adverse events (most commonly neutropenia) occurring in 58% of patients.

There is currently no specific recommended second-line treatment for CCA. In 2021, results of the randomized Phase 3 ABC-06 trial using FOLFOX (in patients who had previously received cisplatin and gemcitabine) showed improved median OS compared to active symptom control alone (6.2 vs. 5.3 months, with Grade 3–5 adverse events reported in 69% of patients in the treatment arm (vs. 52% of those on active symptom control alone) [37]. Yet, so far, despite over 20 studies including 14 Phase 2 clinical trials, there is insufficient data to recommend any chemotherapy regimen over the others based on efficacy [7].

Radiotherapy is another modality of treatment being investigated in patients with locally advanced CCA. Although there may not be a clear survival benefit, chemoradiation was shown to provide effective local disease control over up to 2 years in the majority of patients in a retrospective analysis of 37 patients with unresectable eCCA [38]. The most frequently investigated agent of chemoradiation for CCA is fluorouracil followed by capecitabine. Ongoing studies include the ABC-07 study (EUDRACT 2014-003656-31) which compares stereotactic body radiotherapy (SBRT) in combination with six cycles of cisplatin and gemcitabine vs. the same chemotherapy combination alone in patients with either intra or extrahepatic CCA.

### 3.1. Targeted Therapy

Some of the most well-known pathways in cellular proliferation, survival, and angiogenesis are dysregulated in iCCA. These include inhibitors that have been developed for well-known signaling pathways including the Raf/MEK/ERK, PI3K-AKT, and JAK/STAT pathway. We discuss pertinent targets of each pathway below.

#### 3.1.1. *IDH*1/2

Mutations in isocitrate dehydrogenase (*IDH*) 1 and *IDH*2, proteins responsible for the oxidative decarboxylation of isocitrate to α-ketoglutarate (α-KG), have been observed in 10–23% of patients with iCCAs (see Figure 1), with a majority being female and from non-Asian (less fluke-associated) treatment centers [39,40,41]. Based on a 2019 systematic review of 46 publications, a mutation in this gene was rarely identified (0.8%) in eCCA [39]. The most common mutation was in R132, with the gain of function R132C being the most common mutation in iCCA (followed by R132L, R132G) [39]. Upregulation in *IDH*1/2 leads to accumulation of the oncogenic metabolite (and possible immunosuppressant) D-2-hydroxyglutarate (D2-HG), leading to cellular hypermethylation and epigenetic changes resulting in inhibition of cell differentiation (see Figure 2) [39,42]. Interestingly, there does not appear to be a significant association between *IDH*1 mutation and overall survival (OS), progression-free survival (PFS), and time to progression [39].

In August 2021, the Food and Drug Administration (FDA) approved the mutant-IDH1 inhibitor ivosidenib (brand name Tibsovo) for adults with previously treated, locally advanced, or metastatic CCA with an *IDH*1 mutation. At the same time, the FDA approved the Oncomine Dx Target Test from Life Technologies as a diagnostic aid in selecting patients with the mutation. This came after a Phase 3 multicenter double-blind controlled trial for chemotherapy-refractory *IDH*1-mutated CCA with 185 patients found significantly increased progression-free survival (PFS) (HR 0.37; 95% CI: 0.25, 0.54; *p* < 0.0001) and longer (but not significantly increased) overall survival [43]. The adverse effects reported include fatigue, gastroenterological complaints (e.g., nausea, abdominal pain, diarrhea, decreased appetite), anemia, rash, and more serious side effects (e.g.,Grade 4 hyperbilirubinemia, Grade 3 jaundice cholestatic, QT prolongation, and Grade 3 pleural effusion) in only 2% of patients. A summary of completed clinical trials for targeted therapies for iCCA is provided in Table 1.

Currently, a Phase 1 trial is in progress to investigate the side effects and best dose of gemcitabine and cisplatin when given together with ivosidenib vs.pemigatinib in treating patients with unresectable or metastatic cholangiocarcinoma (NCT04088188). Another Phase 1, multicenter study will determine the safety, pharmacokinetics, pharmacodynamics, and clinical activity of ivosidenib in advanced solid tumors, including glioma and CCA, that harbor an *IDH*1 mutation (NCT02073994). A summary of ongoing clinical trials with targeted therapies is provided in Table 2.

#### 3.1.2. FGFR2 Fusions or Rearrangements

Mutations in the fibroblast growth factor receptor (*FGFR*) 2, including activating translocations (through fusion or rearrangements), are seen in 13–20% of iCCAs (see Figure 1) and detectable by both Next Gen-sequencing as well as FISH [3,44,45]. Fusions and rearrangements upstream of the coding region of *FGFR*2 result in its upregulation and subsequent constitutive activation of growth factor signaling, with increased cell proliferation, metastasis, and angiogenesis (see Figure 3) [44]. These mutations are also rare in eCCAs and more commonly seen in women, but mutations in *IDH* and *FGFR*2 are mutually exclusive, despite an overexpression of *FGFR* 2, 3, and 4 seen in patients carrying *IDH*1/2 mutations [3,44]. Two studies by Graham et al., (with samples from a North American population) and Arai et al., (with samples from a Japanese population) have shown conflicting results regarding the prognostic implications of *FGFR*2 fusions, with the former suggesting a favorable prognosis [44,46].

FGFR inhibitors are generally ATP-competitive, reversible inhibitors (e.g., erdafitinib, infigratinib, pemigatinib, and derazantinib), except for the covalent inhibitor futibatinib [3]. Pemigatinib was approved in April 2020 by the FDA as the first targeted therapy against CCA. This came after the FIGHT-202 trial, a Phase 2 multicenter clinical trial (NCT02924376) spanning North America, Europe, and Asia, that showed a 35.5% objective response rate (3 complete and 35 partial responses) in patients with advanced/metastatic or surgically unresectable CCA with *FGFR*2 translocations who had failed prior therapy [47]. The median duration of response was 9.1 months. The majority (64%) of patients had Grade 3 or more severe adverse events, with the most frequent adverse effect being hypophosphatemia. More serious side effects included cholangitis and pleural effusions (each 3%), but no treatment-related deaths were reported. Following pemigatinib, infigratinib (an FGFR1-3 inhibitor) was approved by the FDA in May 2021 largely based on the success of a Phase 2 trial (NCT02150967), showing an ORR of 23.1% in patients with previously treated advanced CCA, median DOR of 5 months and median PFS of 7.3 months (95% CI, 5.6–7.6 months) [48]. Side effects of infigratinib were similar to those of pemigatinib and included electrolyte abnormalities such as hyperphosphatemia (72%), fatigue, dry eyes, stomatitis, and rare ophthalmic toxicities (e.g., keratitis, trichiasis, and retinal toxicity).

While more selective and potent FGFR2 inhibitors are being developed, current FGFR inhibitors are under further investigation as monotherapies and in combination with chemotherapy. For example, an ongoing Phase 3 study is comparing the efficacy and safety of futibatinib vs. gemcitabine-cisplatin combination as first-line treatment of patients with advanced, metastatic, or recurrent unresectable iCCA with FGFR2 gene rearrangements (NCT04093362). This agent has been suggested as an option for patients who have progressed on previous ATP-competitive inhibitors given its effectiveness against several acquired resistance mutations [3,44]. The PROOF301 trial (NCT03773302) is another Phase 3 trial that uses infigratinib as a first-line therapy compared to the gemcitabine-cisplatin combination. Meanwhile, FIGHT-302 (NCT03656536), compares pemigatinib to chemotherapy as a first-line treatment in advanced cholangiocarcinoma (See Table 2). Additionally, as mentioned previously, a Phase 1 trial is comparing a pemigatinib-gem/cis combination to ivosidenib-gem/cis (NCT04088188). Finally, the FIGHT-101 dose-escalation study to evaluate the safety, tolerability, and pharmacological activity of pemigatinib alone and in combination with chemotherapy such as gemcitabine and cisplatin in subjects with advanced malignancies including CCA is still in process (NCT02393248).

#### 3.1.3. *NTRK* Fusions

Neurotrophic receptor tyrosine kinase (*NTRK*) 1-3 gene fusions are driver mutations that lead to constitutively active kinase signaling. Different fusion proteins (e.g., the product of the fusion gene ETV6-*NTRK*3 or TFG-TRKA) found in different cancers may activate the MAPK and PI3K pathways [49]. By immunohistochemistry and NGS of 145 iCCA and eCCA samples from a hospital in Belgium (CUB HôpitalErasme), *NTRK* fusion is a rare finding seen in 0.75% of biliary tract cancers [50]. In 2018 and 2019, the FDA granted accelerated approval of two inhibitors, entrectinib (a pan-TRK, ROS1, and ALK inhibitor) and larotrectinib for patients with solid tumors with *NTRK* fusions (see Figure 3). This came after the basket Phase 1 trials, STARTRK-1 and ALKA-372-001 demonstrated robust anti-tumor activity (57% response rate) and safety profile (majority of adverse effects were Grades 1 and 2, and most commonly fatigue and dysgeusia) of entrectinib against various solid tumors including in patients with CNS disease [3,51]. Several basket Phase 1 and 2 trials then demonstrated a high overall response rate (75%) to larotrectinib (NCT02122913, NCT02637687, and NCT02576431) [52]. Notably, both have had long durations of response (10 months for entrectinib and not reached for larotrectinib) [3,52]. Ongoing trials include an open-label, multicenter, global Phase 2 basket study of entrectinib (RXDX-101) for the treatment of patients with solid tumors that harbor an *NTRK*1/2/3, *ROS1*, or *ALK*gene fusion (STARTRK-2 or NCT02568267). Another study looks at the efficacy and safety of alectinib in participants with anaplastic lymphoma kinase (ALK)-positive locally advanced or metastatic solid tumors other than lung cancer (ALpha-T or NCT04644315). Additionally, second-generation TRK inhibitors, selitrectinib (LOXO-195) and repotrectinib (TPX-005), are believed to be able to overcome treatment resistance through acquired *NTRK* kinase domain mutations and are the subject of ongoing Phase 1 and 2 basket trials (NCT03215511 and NCT03093116).

#### 3.1.4. *BRAF* V600E

The activating V600E mutation in serine-threonine kinase B-raf kinase (*BRAF*) is found in just 5% of iCCAs. In a Phase 2 trial (ROAR or NCT02034110) completed in June 2022 of da*BRAF*enib used in combination with the MEK inhibitor, trametinib in 43 patients with *BRAF* V600E-mutated, unresectable, metastatic, locally advanced, or recurrent biliary tract cancer the ORR was 47%. A minority of patients (40%) reported serious treatment-related adverse effects, including pyrexia as the most common (19%) effect [53]. These results have led to tissue-agnostic accelerated approval by the FDA for unresectable or metastatic solid tumors that have progressed with no satisfactory alternative options. An ongoing basket Subprotocol H trial (EAY131-H) of the NCI-MATCH platform trial (NCT02465060) with 35 patients with solid tumors, lymphomas, and multiple myeloma with *BRAF* V600E showed an ORR of 38%, with PFS of 11.4 months [54].

#### 3.1.5. *HER2*/ERBB2 Amplifications

Overexpression or amplification of the EGFR family receptor tyrosine-protein kinase erbB-2 is seen in 15–20% of extrahepatic CCAs and gallbladder carcinomas [3]. A basket trial (NCT02091141) with small biliary tract cancer cohorts showed an objective response to combination treatment with trastuzumab and pertuzumab in two of seven patients, while another showed partial response to neratinib in two of nine patients (NCT01953926) [3,55,56]. A multicenter, non-randomized, Phase 2 study to assess the efficacy, safety, and pharmacokinetics of trastuzumab emtansine in participants with *HER2* overexpressing solid tumors including CCA where other treatment options have been exhausted (NCT02999672) was terminated early when five of seven patients with advanced pancreatic/CCA developed progressive disease. Another study (NCT00478140) was terminated due to slow accrual, though it did show that of three patients who received trastuzumab, one had CR while another had stable disease. Several trials have been completed but have not had reported results. These include: (1) a Phase 1 trial to study the effectiveness of interleukin-12 and trastuzumab in treating patients who have cancer that has high levels of *HER2*/neu and has not responded to previous therapy (NCT00004074); and (2) a Phase 1 trial to study the effectiveness of trastuzumab plus R115777 in patients who have advanced or metastatic cancer (NCT00005842). Given the modest effect of *HER2* inhibition as monotherapy, there are currently Phase 1 and Phase 2 trials to determine the safety and efficacy of trastuzumab in combination with other antineoplastic drugs (pembrolizumab, FOLFOX, and CAPOX) (NCT04430738). Another Chinese-based Phase 2 trial aims to determine the efficacy and safety of trastuzumab combined with FOLFIRINOX in advanced or recurrent extrahepatic cholangiocarcinoma and gallbladder carcinoma (NCT03768375). Finally, a Phase 1 study is also underway to characterize the safety and tolerability of the immune checkpoint inhibitor tebotelimab by itself, and in combination with margetuximab (a chimeric IgG monoclonal *HER2* antibody) in *HER2*+ advanced solid tumors (NCT03219268).

Despite the many targeted therapies available, there are unfortunately many non-targetable mutations in CCA which often carry prognostic significance. In iCCA, these include NRAS (17%) and BAP1 (14%). In eCCA these are KRAS (47%), TP53 (24%), and ARID1A (16%).

### 3.2. Immunotherapy

#### 3.2.1. Monotherapy

Immune checkpoint inhibitors (ICIs) are known to potentiate the body’s own adaptive immunity against cancer cells through inhibition of the interaction between positive programmed death-ligand 1 (PD-L1) and its receptor PD-1, or between cytotoxic T lymphocyte-associated antigen-4 (CTLA-4) and its receptors on antigen-presenting cells [57]. They are commonly used in a wide variety of solid tumors with positive programmed death-ligand 1 (PD-L1) expression, high microsatellite instability (MSI-H), mismatch repair (MMR) deficiency, and/or high mutation burden (TMB-H). PD-L1 expression is seen in of 9.1–72.2% of patients with CCA [58].

As one of the most well-known PD-1 inhibitors, pembrolizumab is an FDA-approved treatment for patients with metastatic or inoperable solid tumors with high microsatellite instability (MSI) or mismatch repair (MMR) deficiency. Clinical trials have generally evaluated the efficacy of ICIs as second-or subsequent-line treatment of CCAs. Two studies, KEYNOTE-158 (NCT02628067; Phase 2; n = 104) and KEYNOTE-028 (NCT02054806; Phase 1; n = 24) have used pembrolizumab as second or later-line therapy in advanced biliary tract cancers. All patients in KEYNOTE-028 and 61 in KEYNOTE-158 had PD-L1–positive (membranous PD-L1 expression in ≥1% of the tumor and associated inflammatory cells or positive staining in stroma) tumors [59,60]. Between the two trials, ORR was 5.8% in KEYNOTE-158 and PFS for 2 months while in KEYNOTE-028, ORR was 13.0% (3/23, all PR; 95% CI, 2.8‒33.6%) and PFS was 1.8 months (95% CI, 1.4‒3.7), respectively. Efficacy of pembrolizumab may also vary depending on the level of PD-L1 expression. As shown by Ahn et al., in a study of 125 CCA patients with tumoral PD-L1 positivity (at least 1% expression), those with at least 50% or more of PD-L1 expression had significantly higher overall response rate and disease control [58]. The same KEYNOTE-158, which included other non-colorectal cancers such as endometrial, gastric, and pancreatic cancers in addition to CCA, pembrolizumab was shown to be effective against cancers with high mutation burden (TMB-H) in addition to MSI-high/dMMR tumors [59,61]. A summary of completed clinical trials in iCCA with immunotherapy is provided in Table 3.

Modest effects were also observed with nivolumab in a Phase 2 study of 54 patients (NCT02829918) that showed an ORR of 11% (5 of 46) and a PFS of 3.7 months. All patients who responded had mismatch repair protein-proficient tumors [62]. Even lower response rates for nivolumab were seen in a Japanese study (JapicCTI-153098) with ORR of 3% (1 out of 30 patients), with OS and PFS of 5.2 (90% CI, 4.5–8.7) months and 1.4 (90% CI 1.4–1.4) months, respectively [63]. Notably, patients chosen for nivolumab monotherapy vs. nivolumab along with gem/cis had unresectable or recurrent biliary tract cancer that was refractory or were intolerant to gemcitabine-based treatment regimens. In another Japanese study (NCT01938612), a PD-L1 inhibitor, durvalumab had a similarly low OR of 4.8% and OS of 8.1 (95% CI, 5.6–10.1) months [64].

In general, ICI as monotherapy has been well tolerated. For pembrolizumab, Grade 3–5 treatment-related adverse effects developed in 13.5% of patients in KEYNOTE-158, including one with Grade 5 renal failure [59]. For nivolumab, the most common treatment-related Grade 3 or 4 adverse effects were hyponatremia (3 of 54 (6%)) and increased alkaline phosphatase (2 of 54 (4%)) [62].

#### 3.2.2. Immunotherapy and Chemotherapy

Given the modest benefit of immunotherapy as monotherapy, a few completed and many ongoing studies aim to determine the safety and efficacy of monotherapy vs. combined immuno- and chemotherapy (e.g., gemcitabine and cisplatin) [63]. The combination has been hypothesized to work synergistically, with chemotherapy increasing the immunogenicity of tumor cells by promoting the release of cytokines and damage-associated molecular patterns (DAMPs), and expression of human leukocyte antigen class 1 (HLA1) molecules among other mechanisms [10].

Phase 1 and 2 trials have found promising results with the combination of PD-1 inhibitors and platinum-based chemotherapy. In the Japanese-based JapicCTI-153098, the investigators found the combination of nivolumab and gem/cis to result in an ORR of 37%, along with OS and PFS of 15.4 and 4.2 months [63]. Another study using the combined nivolumab and gem/cis (NCT03311789) has provided promising results with ORR of 55.6% and OS and PFS of 8.5 and 6.1 months atfollow-up of 12.8 months [65].

So far, there are two terminated or completed clinical trials for pembrolizumab used in combination with other agents. The first, a Phase 2 study of just four patients using pembrolizumab and pegylated (PEG) interferon (IFN) alpha-2b (NCT02982720), was terminated in 2019 due to intolerable side effects of PEG-IFN. Another Phase 2 study (NCT03111732), which was completed in November 2021, looked at the combination of pembrolizumab and capecitabine and oxaliplatin (CAPOX) in 11 patients with advanced biliary tract cancer and found no patients with complete response, but 3 (27.3%) with partial response, with 4 patients (36.4%) experiencing a serious side adverse event (most commonly involving the GI system with diarrhea, colitis, and/or ascites) [29,66].

Two other trials assessed the efficacy of camrelizumab combined with gem/cis as first-line therapy against CCA, yielding similar median OS and PFS. The first (NCT03486678), which employed camrelizumab at (3 mg/kg), gemcitabine (800 mg/m^2^), and oxaliplatin (85 mg/m^2^), yielded an impressive ORR of 20 of 37 (54%), which was even higher in patients with positive PD-L1 expression (80% vs. 54.6%). Median PFS and OS were 6.1 and 11.8 months, with themost common Grade 3 or worse TRAEs being hypokalemia (19%) and fatigue (16%) [67]. The second study (NCT03092895) looked at the response of an Asian patient population to combined camrelizumab and either gem/ox (Cam-GEMOX) or FOLFOX (Cam-FOLFOX). Overall ORR of the 92 patients analyzed was 16.3% (95% CI = 9.4–25.5), with 75.0% (95% CI = 64.9–83.4) achieving control of disease. Median PFS and OS were 5.3 months (95% CI = 3.7–5.7) and 12.4 months (95% CI = 8.9–16.1) [68]. Comparing the two groups, Cam-FOLFOX yielded a slightly lower ORR (10.3% vs. 19%), but similar median PFS and OS [10,68].

Another PD-1 inhibitor, toripalimab has been combined with gemcitabine and S-1 (an oral anticancer drug consisting of tegafur, which is a prodrug of 5-FU, combined with gimeracil, and oteracil) in a Phase 2 study of another Asian patient population (NCT03796429). The investigators found an ORR of 27% (13 of 48), mPFS of 7 (95% CI 5.5–9.1) months, and mOS of 16.0 (95% CI 12.1-not reached) [69]. Grade 3/4 side non-hematological TRAEs were seen in 24% of patients and included rash, infection, and colitis. A majority (62%) had Grade 3/4hematological TRAEs and treatment was discontinued in 6 of 48 (12.5%) due to TRAEs.

The PD-L1 inhibitor, durvalumab has also been used in combination with gem/cis (NCT03046862) with promising results. ORR was 73% with mOS and mPFS of 18.1 and 11 months, respectively [70]. Interestingly, when tremelimumab is added to this combination for dual PD-L1 and CTLA4 inhibition, results were similar with ORR of 73.4%, mOS of 20.7 months, and mPFS of 11.9 months.

#### 3.2.3. Dual Immunotherapy

The combination of durvalumab and tremelimumab without chemotherapy has been tested in a Phase 1 trial in patients who had received a median of two prior chemotherapy regimens (NCT01938612), with modest results. ORR was 10.8% vs. 8%, with mOS of 10.1 (95% CI 6.2–11.4) months vs. 8.1 (95% CI, 5.6–10.1) months [64]. Another combination of PD-1 and CTLA4 blockade, nivolumab, and ipilimumab has been tested in a Phase 2 trial (NCT02923934, CA209-538) with 39 BTC (including 33 who had received 1 or more prior lines of treatment) with ORR of 24% (including 3 of 14 of those with iCCA and 1 of 10 in those with eCCA) [71]. None of the responders had microsatellite unstable disease. Median OS and PFS were 6.1 and 3.1 months, with Grade 3/4irAEs observed in 20% of patients.

#### 3.2.4. Immunotherapy and Targeted Therapy

The combination of transforming growth factor (TGF)-beta-blockade and immunotherapy has also been explored through the use of Bintrafuspalpfa (M7824), a bifunctional antibody that targets PD-L1 while linked to an extracellular domain of TGF-beta receptors that trap TGF-beta within the tumor microenvironment. Blocking this immune-suppressive cytokine is hypothesized to prevent tumor progression and metastasis in CCA [10]. A Phase 1 study of Asian patients with CCA who progressed on first-line chemotherapy (NCT02699514) demonstrated an ORR of 20%, mPFS 2.5 (95% CI: 1.3–5.6) months, mOS of 12.7 (95% CI: 6.7–15.7) months, with all except one of the six responders having a DOR of more than 12 months. Another larger study (NCT03833661) of 159 patients who had progressed on first-line chemotherapy has so far shown an ORR of 10.1% (95% CI 5.9–15.8%), mPFS of 1.8 (95% CI 1.7–1.8) months, and mOS of 7.6 (95% CI: 4.8–0.5) months [62].

There are many other combination treatments with pembrolizumab and inhibitors against VEGFR2 (ramucirumab), tyrosine kinases (ramucirumab), and FGFR2 (pemigatinib) as shown in Table 3. Results of the last combination (NCT02393248) are eagerly awaited. For nivolumab, there is a completed Phase 2 trial (NCT03250273) of Entinostat in combination with nivolumab in previously treated unresectable or metastatic CCA and pancreatic adenocarcinoma, which has unfortunately shown a 0% ORR.More ongoing studies are listed in Table 4.

## 4. Current and Future Directions

Despite the advances in CCA management made in the last decade, the prognosis has remained poor for locally advanced and metastatic diseases. Using the ABC-02 trial demonstrating median OS of 11.7 months and PFS of 8 months for the current standard of care first-line systemic therapy as a starting point, a multitude of agents has sought to extend OS and PFS mainly as second-line agents. The discovery of targetable mutations in *IDH*1, *FGFR*2, *HER2*, *BRAF*, and *NTRK*, as well as subgroups with PD-L1 expression and/or MSI-high/TMB-high has revolutionized the field. The most impressive results have been achieved with the combination of gemcitabine, cisplatin, and nab-paclitaxel in (showed a median OS of 19.2 months and PFS of 11.8 months), as well as the combination of durvalumab and gem/cis (NCT03046862) that showed an ORR of 73% with mOS and mPFS of 18.1 and 11 months, respectively, in chemo-naive patients with BTC. For the former, the significant adverse effects associated with the triple chemotherapy regimen limit its widespread implementation. For the latter, it is interesting to note that efficacy was not associated with pre-treatment TMB of the tissue, but rather the PD-L1 expression after the first cycle of gem/cis [70]. As expected, variations in genetic profiles, especially in mutations affecting DNA damage repair, cell cycle regulation, and genome stability also impacted response to treatment.

Notably, based on the KEYNOTE-28 and -158 trials, immunotherapy by itself was generally well-tolerated with only 8% and 6%developing Grade 3 immune-mediated adverse effects and no Grade 4 or 5 effects [3,46]. When combined with chemotherapy, hematological adverse effects such as febrile neutropenia and thrombocytopenia arise as seen in the Phase 1 JapicCTI-153098 trial combining nivolumab with gem/cis [63]. It is unclear if these effects occur at the same frequency and severity as with gem/cis by itself or are worsened by the addition of immunotherapy.

Targeted therapy aside from immunotherapy has offered increased PFS while being generally well-tolerated. Depending on the target, this has ranged from 2.7 (using ivosidenib) to 11.4 months or even unreached at 1-year follow-up (with the combined dabrafenib and trametinib or larotrectinib). The highest response rates and overall survival have been observed with combinations of immunotherapy and chemotherapy. For example, the treatment with durvalumab in combination with gem/cis (NCT03046862) yielded an ORR of 73.3% with OS and PFS of 18.1 and 11 months, respectively. Similar ORR has been seen with the addition of tremelimumab to this combination, with no unexpected adverse effects [70]. In the meantime, the Phase 3 TOPAZ-1 study (NCT03875235) evaluating the combination’s efficacy as first-line therapy has seen a median OS of 12.8 months (compared to 11.5 for gem/cis alone), making it a potential alternative to first-line gem/cis.

Currently, the use of targeted therapy is still limited by the presence of targetable mutations, spurring continuing efforts to discover new targetable mutations, understand mechanisms of resistance, and design better agents which may bind more selectively or potently to their intended targets. More recently, studies have found thattumor-infiltrating lymphocytes (TILs) may also have a role in CCA. In a widely cited paper from 2014, Tran et al. saw disease stabilization in a patient who was treated with TILs (CD4+ T helper 1 cells) that recognized mutated erbb2 interacting protein (ERBB2IP) expressed by metastatic CCA cells [72]. Even after progression, infusion of a purer population of mutation-reactive TILs again induced tumor regression.

Numerous trials are still underway to determine the safety and efficacy of various combinations of chemotherapy, targeted therapy (including immunotherapy),and radiation against CCA.It is not difficult to see the potential in targeted therapies as neoadjuvant or adjuvant therapies with LT, increasing the pool of patients eligible for LT and minimizing recurrence afterwards. Considering the generally greater tolerability of targeted therapy, these treatments may also be preferred in the transplant population.

## 5. Conclusions

While unresectable or metastatic CCA is associated with a much poorer prognosis compared to CCA that meets the criteria for surgical removal or transplant, there is an ever-increasing number of treatment options for the patients with the former type of disease. In clinical trials, certain combination therapies (e.g., triple chemotherapy or immunotherapy with gem/cis combinations) used as first-line treatment have been shown to improve median OS over the standard first-line gem/cis therapy by up to 9 months as seen with the combination of durvalumab, tremelimumab, and gem/cis (NCT03046862). Though the TOPAZ-1 trial showed only a 1.3-month improvement in mOS with durvalumab and gem/cis compared to gem/cis alone, given the lack of significant additional adverse effects, the former is now being considered as an alternative to the standard gem/cis regimen. Further studies will be needed to compare outcomes of combination chemotherapy and immunotherapy vs. sequential treatment. At the same time, given the significant percentage of CCA patients with actionable mutations, approved and experimental targeted therapies are playing a bigger role in disease management. This opens up a whole new set of combination therapies to explore. While systemic therapies are mostly being used in metastatic and unresectable CCA, restaging is often warranted to re-evaluate for resectability and/or liver transplant, which may even further improve outcomes in certain patients. As we investigate combinations of treatment modalities and observe their synergistic anti-tumor action, we may finally tap into the full potential of targeted therapies for CCA.

## Figures and Tables

**Figure 1 cancers-14-05074-f001:**
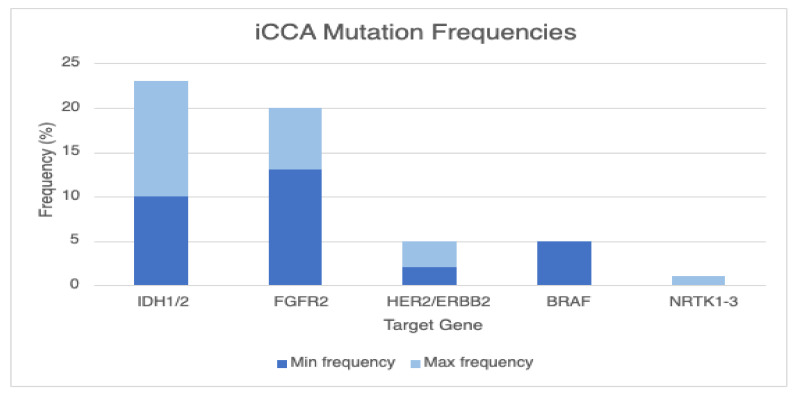
Mutation frequencies in iCCA. From the most common to the least common, ranges of frequencies of targetable mutations in Isocitrate dehydrogenase (*IDH*) 1/2, fibroblast growth factor receptor (*FGFR*)2, *HER2*/ERBB2, B-raf kinase (*BRAF*), and neurotrophic receptor tyrosine kinase (NRTK) 1-3 are provided.

**Figure 2 cancers-14-05074-f002:**
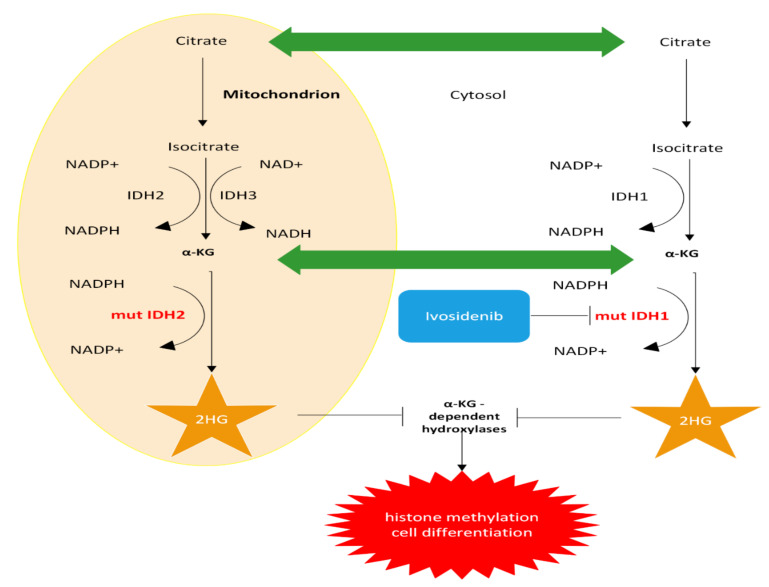
Targeting mutated (mut) isocitrate dehydrogenase (*IDH*) 1 and 2 in iCCA. α-KG: alpha-ketoglutarate; 2HG: 2-hydroxyglutarate.

**Figure 3 cancers-14-05074-f003:**
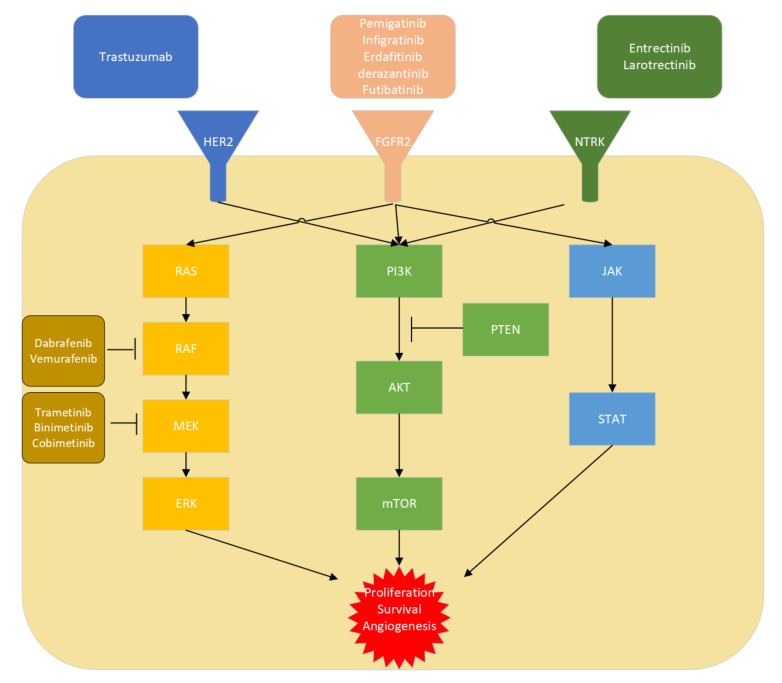
Molecular pathways in iCCA. Inhibitors have been developed for well-known signaling pathways including the Raf/MEK/ERK, PI3K-AKT, and JAK/STAT pathway. AKT: protein kinase B; FGFR: fibroblast growth factor; HER2: epidermal growth factor receptor 2; JAK: janus kinase; mTOR: mammalian target of rapamycin; PI3K: phosphoinositide 3-kinase. NTRK: neurotrophic receptor tyrosine kinase.

**Table 1 cancers-14-05074-t001:** Primary outcomes of completed clinical trials of targeted inhibition in iCCA. ORR: overall response rate. PFS: progression-free survival.

Target	Agent	Clinical Trial	Primary Outcome
IDH1	Ivosidenib	ClarIDHy (NCT02989857), Phase 3	PFS: median 2.7 months [95% CI 1.6–4.2] vs. 1.4 months [1.4–1.6]
FGFR2	Pemigatinib	FIGHT-202 (NCT02924376), Phase 3	ORR: 35.5% [95% CI 26.5–45.4]
	Infigratinib	NCT02150967 (not complete)	ORR: 23.1% [95% CI 15.6–32.2]
MSI-high, MMR-deficient, or TMB-H	Pembrolizumab	KEYNOTE-028 (NCT02054806), Phase 1b, PDL-1 positive tumors only	ORR: 13.0% (3/23; 95% CI, 2.8–33.6%)
		KEYNOTE-158 (NCT02628067), Phase 2 (not complete)	ORR: 5.8% (6/104; 95% CI, 2.1–12.1%)
			ORR of PD-L1-positive: 6.6% (95% CI: 1.8–15.9%)
			Basket TMB-H group: ORR: 29% (95% CI 21–39)
			CCA subgroup ORR: 40.9% (95% CI, 20.7–63.6%)
BRAF	dabrafenib and trametinib	ROAR trial (NCT 02034110), Phase 2	ORR: 47% (20/43; 95% CI 36–67%)
		Subprotocol H trial (EAY131-H)	ORR: 38% (90% CI, 22.9% to 54.9%)
NTRK	Entrectinib *	STARTRK-1(NCT02097810) (basket Phase 1)	ORR: 57% (31/54; 95% CI 43.2–70.8%)
		STARTRK-2 (NCT02568267) (basket Phase 2)	
	Larotrectinib *	NCT02122913, NCT02637687, and NCT02576431	ORR: 75% (95% confidence interval [CI], 61 to 85)

* Not approved for CCA but recommended by NCCN as first or subsequent-line treatment.

**Table 2 cancers-14-05074-t002:** Ongoing targeted therapy trials in CCA.

Target	Agents	Clinical Trial	Notes
**IDH1**	Ivosidenib + Gemcitabine + cisplatin	NCT04088188, phase I	
	Ivosidenib	NCT02073994	
	LY3410738 ± (Gemcitabine + cisplatin) or durvalumab	NCT04521686, phase I	LY3410738 is a potent, selective, and covalent inhibitor of IDH1-R132
	AG-120 (oral Ivosidenib)	NCT02073994, phase I	
	IDH305	NCT02381886, phase I	For IDH1R132-mutant tumors only
**FGFR2**	oral infigratinib or gemcitabine + cisplatin	NCT03773302, phase III	
	futibatinib or gemcitabine + cisplatin	NCT04093362 phase III	
	derazantinib	NCT03230318, phase II	Derazantinib is a potent FGFR1‒3 kinase inhibitor
	atezolizumab + derazantinib	NCT03230318 *, phase II	
	E7090	NCT04238715 *, phase II	E7090 is a selective tyrosine kinase inhibitor against FGFR1-3
	Pemigatinib or Gemcitabine + Cisplatin	NCT03656536 (FIGHT-302), phase III	
	RLY-4008	NCT04526106 (REFOCUS), phase I and II	RLY-4008 is a potent and highly selective FGFR2 inhibitor
	KIN-3248	NCT05242822, phase I	KIN-3248 is an oral small molecule FGFR inhibitor
	Futibatinib	NCT02052778, phase I and II	
	Bemarituzumab	NCT05325866, phase I	Bemarituzumab (FPA144) is a humanized, afucosylated immunoglobulin G1 monoclonal antibody (mAb) against FGFR2b
**BRAF**	ABM-1310	NCT05501912 *, phase I	
	ABM-1310 ± cobimetinib	NCT04190628, phase I	Cobimetinib is a MEK inhibitor approved for the treatment of advanced melanoma
**NTRK**	Entrectinib	NCT02568267, phase I	Entrectinib is a pan-TrkA/B/C, ROS1, and ALK inhibitor
**HER2**	tucatinib + trastuzumab ± (pembrolizumab or FOLFOX or CAPOX)	NCT04430738, phase I and II	tucatinib is a selective tyrosine kinase inhibitor of HER2
	A166	NCT03602079, phase I and II	A166 is an antibody-drug conjugate composed of a novel cytotoxic drug (Duo-5, anti-microtubule agent)
	CT-0508	NCT04660929, phase I	CT-0508 is a biologic composed of adenovirally-transduced autologous macrophages containin Anti-HER2 Chimeric Antigen Receptor (CAR macrophages)
	tebotelimab ± margetuximab	NCT03219268, phase I	Tebotelimab is a DART molecule designed to bind PD-1 and LAG-3. Margetuximab is an anti-HER2 antibody.
	Zanidatamab	NCT04466891 (HERIZON-BTC-01), phase II	
	Zanidatamab + XELOX or (FOLFOX ± bevacizumab) or (cisplatin + Gemcitabine or 5-FU)	NCT03929666	
	DB-1303	NCT05150691, phase I and II	DB-1303 is an antibody-drug conjugate

* Non-US-based study.

**Table 3 cancers-14-05074-t003:** Completed immunotherapy clinical trials in CCA.

Monotherapy				
Target	Agent	Clinical Trial	Response Rate	OS and PFS
PD-1, MSI-high, MMR-deficient, or TMB-H	Pembrolizumab	KEYNOTE-028 (NCT02054806), Phase 1b, PDL-1 positive tumors only	ORR: 13.0% (3/23; 95% CI, 2.8–33.6%)	OS: 5.7(95% CI, 3.1–9.8) months; PFS: 1.8 (95% CI, 1.4–3.1) months
	Nivolumab	NCT02829918	ORR: 11%	OS: 14.2 (95% CI, 5.98 to not reached) months; PFS 3.68 (95% CI, 2.30–5.69) months
		JapicCTI-153098	ORR: 3%	OS: 5.2 (90% CI, 4.5–8.7) months; PFS 1.4 (90% CI 1.4–1.4) months
PD-L1	Durvalumab	NCT01938612	ORR: 4.8%	OS: 8.1 (95% CI, 5.6–10.1) months; PFS 2 months
**Dual** **Immunotherapy**				
PD-L1 and CTLA4	Nivolumab and Ipilimumab	CA209-538	ORR: 23%	OS: 5.7 months; PFS 2.9 months
	Durvalumab and Tremelimumab	NCT01938612	ORR 10.8%	OS: 10.1 months
PD-L1 and TGF-β RII	Bintrafusp alfa	NCT02699514	ORR: 20%	OS: 12.5 months; PFS 2.5 months
		NCT03833661	ORR: 10.1%	PFS 1.8 (95% CI 1.7–1.8) months, OS of 7.6 (95% CI: 4.8–0.5) months.
PD-1 and	PEG-intron	NCT02982720	Terminated early	
**ICI and chemo**				
PD-1	Nivolumab and gem/cis	JapicCTI-153098	ORR: 37%	OS: 15.4 months; PFS 4.2 months
		NCT03311789	ORR 55.6%	OS: 8.5 months; PFS 6.1 months
	Camrelizumab, gemcitabine/oxaliplatin	NCT03486678	ORR: 54%	OS: 11.8 months; PFS 6.1 months
	Camrelizumab, gemcitabine/oxaliplatin or FOLFOX	NCT03092895	ORR: 16.3%	OS: 12.4 months; PFS 5.3 months
	Toripalimab, gemcitabine/S-1	NCT03796429	ORR: 27%	OS: 16 months; PFS 7 months
	Pembrolizumab and CAPOX	NCT03111732	ORR: 27.3%	OS: 6.1 (95% CI 3.1–9.4); PFS: 4.5 (95% CI 2.5 to 9.6) months
PD-L1	Durvalumab, gem/cis	NCT03046862	ORR: 73.3%	OS: 18.1 months; PFS 11 months
CTLA4	PEGCISGEM ± Atezolizumab	NCT03267940	AEs: 100% in both	none (early termination)
PD-L1 and CTLA4	Durvalumab + gem/cis ± tremelimumab	NCT03046862	ORR: 73.4%	OS: 20.7 months; PFS 11.9 months
**Other combinations**				
PD-1 and VEGFR2	Pembrolizumab and Ramucirumab	NCT02443324	ORR: 4%	OS: 6.4 months; PFS 1.6 months
PD-1 and tyrosine kinase	Pembrolizumab/nivolumab and Ramucirumab	NCT03892577	ORR: 30%	OS: 11 months; PFS 5 months
PD-1 and FGFR2	Pembrolizumab and pemigatinib (dose escalation)	NCT02393248	-	
PD-1 and benzamide histone deacetylase	Nivolimumab and Entinostat	NCT03250273	ORR: 0%	OS: 6.378 (95% CI 3.748-NR at 36 months) months
PD-1 and cell-based	Pembrolizumab and CD8+ T-cells	NCT02757391	-	
PD-L1	Durvalumab, tremelimumab, radiotherapy	NCT03482102	ORR: 25%	

**Table 4 cancers-14-05074-t004:** Ongoing Immunotherapy-based Trials in CCA.

Immunotherapy	ClinicalTrials.gov ID	Agents Used	Study Phase	Number Enrolled
Pembrolizumab	NCT04550624 *	Pembrolizumab + Lenvatinib	Phase 2	40
	NCT04306367	Pembrolizumab + Olaparib	Phase 2	29
	NCT05220722	Pembrolizumab/Nivolumab/Ipilimumab + Pressure Enabled Delivery of SD-101	Phase 1, 2	89
	NCT03895970 *	Pembrolizumab + Lenvatinib	Phase 2	50
	NCT03781934 *	Pembrolizumab/Lenvatinib + MIV-818	Phase 1, 2	102
	NCT02628067	Pembrolizumab (MK-3475-158/KEYNOTE-158)	Phase 2	1595
	NCT04430738	(Tucatinib + trastuzumab) ± pembrolizumab ± (FOLFOX or CAPOX) in HER2+ cancers	Phase 1, 2	120
	NCT05215574	NGM831 ± Pembrolizumab	Phase 1	79
	NCT04913337	NGM707 ± Pembrolizumab	Phase 1, 2	179
	NCT03849469	XmAb^®^22841 ± Pembrolizumab	Phase 1	242
	NCT05007106	Pembrolizumab ± Vibostolimab ± (5-Fluorouracil + Cisplatin) ± Paclitaxel	Phase 2	480
	NCT03329950	CDX-1140 (CD40) ± (Pembrolizumab or CDX-301)	Phase 1	260
	NCT03058289	Intratumorally Dosed INT230-6 (cisplatin and vinblastine sulfate) ± (pembrolizumab or ipilimumab)	Phase 1, 2	180
	NCT03872947	TRK-950 + (Gemcitabine + Cisplatin or Pembrolizumab)	Phase 1	181
	NCT04924062 *	Gemcitabine/Cisplatin ± Pembrolizumab (MK-3475-966/KEYNOTE-966)-China Extension Study	Phase 3	160
	NCT04003636	Gemcitabine/Cisplatin ± Pembrolizumab (MK-3475) (MK-3475-966/KEYNOTE-966)	Phase 3	1048
	NCT04157985	PD-1/PD-L1 Inhibitors (pembrolizu/nivolimu/atezolimu/ipilimu/cemiplimab) 1-year vs. more	Phase 3	578
Nivolumab	NCT05220722	Pembrolizumab/Nivolumab/Ipilimumab + Pressure Enabled Delivery of SD-101	Phase 1, 2	89
	NCT03829436	TPST-1120 ± Nivolumab	Phase 1	138
	NCT02834013	Nivolumab ± Ipilimumab	Phase 2	818
	NCT03684811	FT 2102 ± Nivolumab ± (Gemcitabine + Cisplatin)	Phase 1|Phase 2	200
	NCT04648319 *	Nivolumab + SBRT	Phase 2	40
	NCT03872947	TRK-950 + (Gemcitabine + Cisplatin) or Nivolumab or Pembrolizumab	Phase 1	181
	NCT04157985	PD-1/PD-L1 Inhibitors (pembrolizu/nivolimu/atezolimu/ipilimu/cemiplimab) 1-year vs. more	Phase 3	578
Durvalumab	NCT04301778	Durvalumab + CSF-1R Inhibitor (SNDX-6532)	Phase 2	30
	NCT04989218	Durvalumab + Tremelimumab + Gem + Cis	Phase 1|Phase 2	20
	NCT04238637 *	Y-90 SIRT + Durvalimumab ± Tremelimumab	Phase 2	50
	NCT03991832 *	Olaparib + Durvalumab in IDH-Mutated Solid tumors	Phase 2	78
	NCT04298008 *	AZD6738 + Durvalumab	Phase 2	26
	NCT04308174 *	Neoadjuvant Gem + Cis ± Durvalumab in Resectable Biliary Tract Cancer	Phase 2	45
	NCT03257761	Guadecitabine + Durvalumab	Phase 1	55
	NCT04298021 *	AZD6738 + Durvalumab or Olaparib	Phase 2	74
	NCT03473574 *	Durvalumab + Tremelimumab + Gemcitabine ± Cisplatin	Phase 2	128
	NCT02821754	Durvalumab + Tremelimumab ± TACE or RFA or Cryo	Phase 2	53
Atezolizumab	NCT05174650	Atezolizumab + Derazantinib	Phase 2	37
	NCT03201458	Atezolizumab ± Cobimetinib	Phase 2	76
	NCT05211323	Gem + Cs ± (Bevacizumab + Atezolizumab)	Phase 2	88
	NCT04708067	Hypofractionated Radiation Therapy + Bintrafusp Alfa	Phase 1	15
	NCT04941287	Atezolizumab + CDX-1127 (Varlilumab) ± Cobimetinib	Phase 2	64
	NCT05000294	Atezolizumab + Tivozanib	Phase 1|Phase 2	29
	NCT04989218	Durvalumab + Tremelimumabwith Platinum-based Chemotherapy in Intrahepatic Cholangiocarcinoma (ICC)	Phase 1|Phase 2	20
	NCT04157985	PD-1/PD-L1 Inhibitors (pembrolizu/nivolimu/atezolimu/ipilimu/cemiplimab) 1-year vs. more	Phase 3	578

* Study conducted outside the United States.

## Data Availability

All supporting data are contained within the sources cited within this article and through publically available data reported on clinicaltrials.gov.

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
