# Peer review of "Newest Therapies for Cholangiocarcinoma: An Updated Overview of Approved Treatments with Transplant Oncology Vision"

_cancers, 2022, doi:10.3390/cancers14205074_

Round 1
Reviewer 1 Report
The manuscript by Zhang et al. constitutes a timely review article on the newest therapies for cholangiocarcinoma. The authors herein provided an excellent review on the topic, highlighting the most relevant findings so far, providing their view mostly related with surgical procedures, such as transplant and tumor resection. The manuscript is very well-written and contains the most up-to-date clinical studies. I only have small concerns that might be addressed before publication.
1 – The authors don’t discuss the use of adjuvant and neoadjuvant therapy for CCA. They should consider to include some information on this concern.
2 – In page 3, first paragraph, there is a reference that is not included as a number. Please revise
3 – All gene names should appear in italics
4 – In page 9, second paragraph, some sentences have a different font size. The same in future perspectives. Please carefully revise all the manuscript.
5 – It would be important to include also a table with the ongoing clinical trials for targetd therapies (at least the most important ones).
6 – Since the authors may want to correlate the use of these therapies with liver transplantion and/or surgery, the author may want to provide their critical opinion on how these therapies may impact the use of surgical procedures in the future.
Author Response
Thank you for your helpful suggestions. We have made the revisions as suggested below.
Point by point response to reviewer:
1 – The authors don’t discuss the use of adjuvant and neoadjuvant therapy for CCA. They should consider to include some information on this concern.
Response: A paragraph discussing adjuvant and neoadjuvant therapies for CCA is included in page 4 under "Systemic therapies".
2 – In page 3, first paragraph, there is a reference that is not included as a number. Please revise.
Response: This has now been addressed in the manuscript.
3 – All gene names should appear in italics
Response: Gene names have been italicized.
4 – In page 9, second paragraph, some sentences have a different font size. The same in future perspectives. Please carefully revise all the manuscript.
Response: Font sizes have been made consistent throughout the manuscript.
5 – It would be important to include also a table with the ongoing clinical trials for targetd therapies (at least the most important ones).
Response: This is now included as Table 2.
6 – Since the authors may want to correlate the use of these therapies with liver transplantion and/or surgery, the author may want to provide their critical opinion on how these therapies may impact the use of surgical procedures in the future.
Response: A discussion has been added under "Current and Future Directions" on page 18.
Thank you.
Reviewer 2 Report
In this article, the authors provide a very detailed overview of different treatment strategies for cholangiocarcinoma. First the authors introduce the disease, its location and biology. Next the authors describe the surgical options for localized disease. This is followed by review of different mutations that are prominently found in CAA. Next the authors described targeted therapies against these mutations. The article then focusses on immunotherapies for CAA and immune-chemo combos. The text is supported by 3 Figures, 3 comprehensive tables and 68 updated references.
The introductory paragraph provides a description of the anatomy, location and histological features of the disease. This paragraph will definitely benefit by addition of a figure describing the different types of CAA and their location.
Introduction section can also benefit from the addition of a Table on staging of the disease.
Font size is different in the Current and Future Directions section
Minor typos were found that should be checked and corrected
Author Response
The authors thank the reviewer for his/her comments and helpful suggestions. Please see the following point-by-point response:
1- The introductory paragraph provides a description of the anatomy, location and histological features of the disease. This paragraph will definitely benefit by addition of a figure describing the different types of CAA and their location.
Response: As the article focuses on the most up-to-date treatments of CCA, we did not think it was necessary to include additional figures on the types of CCA as these have been discussed by previously published reviews and articles on CCA characterization.
2- Introduction section can also benefit from the addition of a Table on staging of the disease.
Response: As the article focuses on the most up-to-date treatments of CCA, which already makes it a fairly lengthy article, we did not think it was necessary to include additional figures on staging which can be accessed from previously published guidelines.
3- Font size is different in the Current and Future Directions section
Response: This has been addressed in the new submission.
4- Minor typos were found that should be checked and corrected
Response: This has been addressed in the new submission.
Thank you.
Reviewer 3 Report
I have made a few minor changes and/or recommendations per comments, which I have sent to the editors.

Author Response
Thank you for your review.